# Pharmacists and Pharmacy Technicians’ Perceptions of Scopes of Practice Employing Agency Theory in the Management of Minor Ailments in Central Indonesian Community Pharmacies: A Qualitative Study

**DOI:** 10.3390/pharmacy11050132

**Published:** 2023-08-22

**Authors:** Vinci Mizranita, Jeffery David Hughes, Bruce Sunderland, Tin Fei Sim

**Affiliations:** 1Department of Pharmacy, Universitas Sebelas Maret, JL. Ir. Sutami No. 36A Kentingan, Surakarta 57126, Indonesia; 2Pharmacy, Curtin Medical School, Curtin University, Bentley, Perth, WA 6102, Australia; j.d.hughes@curtin.edu.au (J.D.H.); b.sunderland@curtin.edu.au (B.S.); t.sim@curtin.edu.au (T.F.S.)

**Keywords:** community pharmacy, minor ailments, community pharmacist, pharmacy technician, agency theory, thematic analysis, Indonesia

## Abstract

Community pharmacy staff assist in the management of minor ailments. Agency Theory underpins relationships between health professionals and patients. This study explores pharmacists’ and pharmacy technicians’ perceived scopes of practice of minor ailment services in community pharmacies. Twelve one-on-one semi-structured interviews used an open-ended interview guide for each cohort of community pharmacists and pharmacy technicians, between June and July 2021. Purposive sampling selected a diversity of pharmacists and pharmacy technicians. Interviews were transcribed verbatim, thematically analysed assisted by NVivo version 20. Agency Theory aided the interpretation. Three main themes emerged: (1) inconsistencies in practice, (2) the lack of understanding of the scopes of practice of pharmacists and pharmacy technicians, and (3) provision of prescription-only medicines for some minor ailments or to fulfil patient requests. Several sub-themes included pharmacy staff involvement, education and training, provision of prescription-only medicines, and weak regulatory enforcement. Agency Theory indicated pharmacy patients (principals) delegated authority to pharmacists and pharmacy technicians (agents), which was confused by partial pharmacist absence. The lack of defined scopes of practice for pharmacists and pharmacy technicians disrupted established professional relationships. The scopes of practice and roles of the pharmacist and pharmacy technicians should be clearly defined, assisted by practice guidelines.

## 1. Introduction

Minor ailments are defined as self-limiting conditions that require little or no medical attention [1]. Successful management of minor ailments (MMA) includes the appropriate selection of non-prescription medicines, providing appropriate dosage and duration information, and appropriate advice and recommendations [2].

Both community pharmacists and pharmacy technicians have roles in the assessment and treatment of specific minor ailments [3,4,5,6,7,8,9]. One of the challenges in the provision of MMA services in community pharmacy in developing countries relates to the differential roles of pharmacists and other pharmacy staff (pharmacy technicians, drug dispensers, etc.) [3,4,5,6,7,8,9,10]. Common and straightforward ailments (e.g., simple headache) may be adequately managed by a pharmacy technician or even trained staff, while other ailments require a pharmacist’s intervention and advice, or referral to other healthcare professionals [11,12]. Pharmacy technicians may not be able to evaluate contextual information that could differentiate a self-limiting condition from one which is more serious and requires additional intervention. In several Asian countries, pharmacy technicians were usurping the pharmacist’s role in providing advice and recommending treatment, rendering the roles of these two pharmacy professionals ambiguous [11,12,13].

In Indonesia, a transformation towards promoting responsible self-medication is occurring to enhance pharmacy services in the MMA [14,15]. However, this change has been slow owing to inconsistent practice and limited enforcement of regulations. The MMA services in Indonesia occur without established guidelines for pharmacists or pharmacy technicians, leading to variable practices in community pharmacies [11,16,17]. Community pharmacies in Indonesia deliver MMA services through qualified and unqualified staff; however, pharmacy technicians and pharmacists are registered practitioners and have expertise for their provision [18,19]. Indonesian pharmacy technicians may perform several pharmacy services under the supervision of a registered pharmacist, but not provide pharmacist-only medicines, unless a pharmacist is involved in their direct supervision. To become a registered pharmacist in Indonesia, one must undertake a minimum of five years of pharmacy education (a four-year Bachelor of Pharmacy (B.Pharm) and a one-year Apothecary degree). To become a registered pharmacy technician, pharmacy technician students must undertake a three-year Pharmacy Diploma after three years of secondary education.

Agency Theory is a model that describes the delegation of decision-making authority by a principal (the patient) to agents (healthcare professionals) to perform activities on the principals’ behalf and the pathway of interactions between the health professionals in this context. This allows interpretation of patient-professional relationships where the patient relies on the expertise of the professional to act responsibly on their behalf, to achieve their desired outcomes. It is extensively used in the economics sector. In pharmacy practice, Agency Theory has been applied to examine pharmacy clients’ attitudes toward the pharmacist’s role in prescribing in therapeutic areas independent of doctors prescribing [20]. That study has shown a patient’s (principal) acceptance of a pharmacist being a secondary prescriber (agent), if a doctor had made the initial diagnosis [20].

Despite the extensive application of Agency Theory in other sectors, there is no published research exploring the scopes of practice of pharmacists’ and pharmacy technicians’ within the context of Agency Theory in the delivery of minor ailments services in Indonesia. This study aimed to explore the underlying factors influencing pharmacists’ and pharmacy technicians’ perceived scopes of practice to professionally deliver minor ailment services in community pharmacies, within the context of Agency Theory, in Central Java, Indonesia.

## 2. Materials and Methods

### 2.1. Ethical Approval

The study was approved by Curtin University, Human Research Ethics Committee, Australia, with approval number: HRE2019-0803-12 (10 June 2021); the IAI Central Java Regional Board, Indonesia with approval number B1-064/PD-IAI/Jawa-Tengah/IX/2019; and PAFI Central Java Regional Board, Indonesia, with approval number 268/PAFI-JTG/XI/2019.

### 2.2. Research Setting

A qualitative interview study was conducted with qualified staff of community pharmacies in Central Java, Indonesia. A maximum variation purposeful sampling method was used in assembling a sample that ensured a broad range of community pharmacy operations in this study. This method was used to obtain pharmacists’ and pharmacy technicians’ views, ensure diversity in pharmacy staff demographics and characteristics, and provide depth to the data to achieve variation in the sample [21]. COREQ (COnsolidated criteria for REporting Qualitative research) checklist was used to guide the conduct and report of the study (Appendix A) [22].

### 2.3. Participant Selection

The inclusion criteria were pharmacists and pharmacy technicians practising in a range of independent/supermarket/co-located medical centre (with direct access for the public to a pharmacy) community pharmacies in various cities based on a range of population sizes, type of pharmacy and location of pharmacy in Central Java, Indonesia. Interviewees were also selected on the basis of years of experience and gender. The exclusion criteria were pharmacists and pharmacy technicians working in a pharmacy located within a clinic or located within a doctor’s clinic (with no direct patient access to the pharmacy), or pharmacies located in a medical skincare clinic.

The local Indonesian Pharmacists Association (Ikatan Apoteker Indonesia—IAI) and Indonesian Pharmacy Technicians Association (Persatuan Ahli Farmasi Indonesia—PAFI) in Central Java, Indonesia were contacted to obtain lists of pharmacists and pharmacy technicians practising in community pharmacies in their localities. The investigator used the lists (e.g., pharmacy type, respondent’s work position) to select the participants purposively based on the above criteria. Pharmacists and pharmacy technicians were selected from different community pharmacies (not the same pharmacy). The investigator contacted the participants between June–July 2021 by telephone and explained about the study. If the participant agreed to participate, a time for the online interview was arranged. If the participant declined, another suitable replacement was selected from the list and was contacted. The interviews were conducted sequentially, analysed, and new participants were recruited until data saturation was ascertained.

### 2.4. Data Collection

The interviews were conducted online due to the COVID-19 pandemic as travel from the university to community pharmacies or visiting households in Indonesia was banned. The investigator conducted the interviews from the university, and the interviewees were respondents residing in several cities in Central Java, Indonesia. These interviews were conducted virtually (Zoom platform, Zoom Video Communications, San Jose, CA, USA, Version 5.15.7 (20303)) due to pandemic-related travel restrictions, and were performed between June-July 2021 at a convenient time (either at the respondents’ workplace or home). A participant information sheet was provided and participants signed a consent form. Demographic data of each respondent were collected online through Qualtrics^®^. The investigator (a female pharmacist with advanced clinical pharmacy qualifications), whose first language is Bahasa Indonesia, conducted all interviews in Bahasa Indonesia using a semi-structured interview guide. The interviews lasted between 45 to 60 min, and were audio- and video-recorded. A gift card was offered for the participants as a token of appreciation ($AUD 10 = Rp. 100,000).

### 2.5. Interview Guides

Two semi-structured interview guides for community pharmacists and pharmacy technicians that consisted of a series of open-ended questions were developed based on each key theme of interest: their perspectives of the MMA, including scopes of practice suitable for community pharmacists and community pharmacy technicians. The relationship between pharmacists and pharmacy technicians in delivering minor ailment services was developed post hoc from the perspective and application of the Agency Theory. These relationships were developed from the researchers’ knowledge, interview descriptions and the selected literature [14,23,24,25,26,27]. This was subsequently applied post hoc to the results of the study (Figure 1). Agency Theory was used to examine the potential for disruption to the established relationships of stakeholders involved in the MMA in community pharmacies in Indonesia. Topics included were iteratively revised for clarity and relevance. The guides were validated by three academics with community pharmacy experience, and three pharmacists (for the pharmacist guide) and three technicians (for the pharmacy technician guide) practising in different types and sizes in community pharmacies in Central Java, Indonesia. Interview guide pre-tests were conducted with a practicing pharmacist and a pharmacy technician who had the same characteristics as the participants and were not included in the study. The trials resulted in minor changes in the interview guides. The interview guides are included in Appendix B.

### 2.6. Data Analysis

The video interviews were transcribed verbatim in Indonesian Bahasa with a summary of the transcription also back-translated to English by the primary researcher (VM). The data for pharmacists and pharmacy technicians were analysed separately in Bahasa Indonesia using inductive thematic analysis through initial reading and re-reading of the transcripts to become familiar with the data, followed by initial coding, identification of sub-themes, and the development of a coding framework [28]. One female pharmacy academic as a second person (from an East Java University, Indonesia) with excellent Bahasa Indonesia and English language skills reviewed the initial data and the themes and provided feedback to provide external validity and trustworthiness of the data [28]. VM conducted the initial coding, and crosschecks and thematic analysis were confirmed by a second member of the research team (TFS). Examples of comments were selected to illustrate each theme and an independent translator and proof-reader was engaged for independent translation of each comment into English. The theme labels and the analysis process were performed separately for pharmacist and pharmacy technician respondents but were combined for the final report.

Data analysis was managed using NVivo (QSR NVivo version 20, QSR International) to organize the qualitative data and quotations. This simplified the identification of themes and codes patterns. Data collection and data analysis were conducted simultaneously throughout the study to identify the point of data saturation. The respondents’ anonymity was protected by replacing their identifiers with a de-identified code (pharmacists become “Pharmxx” and pharmacy technicians become “Techxx”).

The Agency Theory framework was used to explain the key themes of interest to align with the relationships of principals’ and agents’ roles in the provision of minor ailment services [29,30]. A model of the Agency Theory was derived from the minor ailments services processes employed in community pharmacy practice (Figure 1). Pharmacist and pharmacy technician respondents were asked about their characteristics, including age, gender, working hours, and educational qualifications. These characteristics were then combined with the thematic analyses of the interviews using the key findings of interest. The Agency Theory was applied when both respondent cohorts discussed their delivery of MMA services and their context within the legally defined practice in the community pharmacy setting in Indonesia [29,31].

## 3. Results

A total of 12 pharmacist and 12 pharmacy technician respondents were interviewed. Data saturation occurred with no new themes emerging, following the ninth pharmacist and similarly for pharmacy technician interviews. A further three pharmacist and pharmacy technician respondents were recruited and interviewed, with these additional three respondents from each group further confirming data saturation. One pharmacist and two pharmacy technicians withdrew from the study (before the interview) due to COVID-19 infections or were taking care of family with COVID-19. One pharmacist declined to participate in the interview due to other commitments and one pharmacist was excluded because their pharmacy was located within a doctor’s clinic. To maintain the initial number, these respondents were replaced.

A majority of the pharmacist and pharmacy technician respondents were female, aged between 21–40 years and had practised between 2 and 10 years (Table 1 and Table 2). Independent pharmacies were the majority type of pharmacy business reported in this study by both respondent groups. Most pharmacies had, on average, between 351 and 700 consumers visiting per week, including more than 70 patients seeking advice for MMA. 

Three main themes emerged in relation to the respondents’ perspectives of the MMA: (1) inconsistencies in MMA practice; (2) lack of understanding of the scopes of practice of pharmacists and pharmacy technicians; and (3) provision of prescription-only medicines for MMA, as shown in Table 3.

### 3.1. Theme One: Inconsistencies in MMA Practice

A number of sub-themes encompassing the theme of inconsistencies in MMA practice were evident. No standard procedures were described from the interviews for providing minor ailment services, unclear task decision and decision making was often evident, absence of qualified pharmacists from the premises during opening hours, non-qualified assistants involvement in providing basic minor ailments services, and inconsistent training and education were major issues that were identified from the interviews.

#### 3.1.1. Pharmacists’ Absence and Lack of Standard Procedures for the Provision of Minor Ailment Services

Inconsistency in practice for minor ailments emerged partially resulting from the pharmacist’s absence from the premises during some of the opening hours:


*“…many people come with a complaint of minor ailments to the pharmacy every day. However, the main obstacle is that the pharmacist is not always at service at the pharmacy. It is also impossible for me to work a full-time shift at the pharmacy…”*
(Pharm10)


*“…In the afternoon, when the pharmacists are not in practice, many patients require self-medication, and we (the pharmacy technicians) have to serve them all, not the pharmacists because they are not in service…”*
(Tech6)

Pharmacists’ absenteeism during pharmacy opening hours was also evident from their average attendance hours (Table 1). Further, the pharmacy technician interviews indicated that pharmacists’ interactions with patients was small and often these activities were inappropriately delegated to other pharmacy staff, either pharmacy technicians or non-qualified assistants. The absence of pharmacists limited their ability to assume any consistent responsibility for MMA provision, which was often highlighted by respondents in this study.

When the pharmacist was absent, patients requesting to consult with a pharmacist would then have to decide whether a pharmacy technician or a non-qualified staff member had the skills to manage their minor ailment. Both respondent groups indicated that all pharmacy staff, regardless of their qualifications, initially assisted customers presenting to the pharmacy and would provide minor ailments services unless the request required higher-level input. However, in the majority of cases, no clear delegation pathways were evident. Further, there was no real interest to change the current pharmacy model: 


*“Sometimes, I am in charge of serving patients (initially). Other times, it can be my wife (a pharmacist), the administrator, and staff (who graduated from pharmacy vocational high school or senior high school). Basically, anyone working in the pharmacy can be in charge of serving patients.”*
(Pharm6)


*“SPG (sales promotion staff) will directly serve patients at the front door, welcome incoming patients, and offer their products to patients.”*
(Tech11)

Both respondent groups reported unclear task delegation and decision-making if a pharmacist was absent from the pharmacy. Pharmacist respondents highlighted variable task delegation when pharmacists were not available at a pharmacy:


*“I always inform pharmacy technicians that whenever they are having certain issues, they can contact me (by phone or WhatsApp) because serving medicine cannot be done carelessly…”*
(Pharm10)


*“If one pharmacist was having a day off, the other pharmacist will only serve in the afternoon shift…in case we need to ask anything, we can consult with a senior staff… we can call the pharmacist to consult the patient’s complaint according to their need.”*
(Tech11)

A pharmacy technician stated her confidence to manage minor ailments and pharmacist involvement was not required:


*“The pharmacist here is freshly graduated and does not have full-time work at the pharmacy, so I never recommend consulting a pharmacist for patients….the pharmacist will consult me because I have worked in this field for nine years…”*
(Tech3)

In the interviews, both respondent groups indicated interrelations between pharmacy staff and the patients. In these interrelations, pharmacy staff (pharmacists and pharmacy technicians) establish the patients’ trust when seeking advice for minor ailments. Pharmacy technicians reported that they referred patients to the pharmacist when the conditions required further assessment.

#### 3.1.2. Non-Qualified Assistants’ Involvement Providing Basic Services

The perceptions from pharmacist and pharmacy technician interviews showed there was an involvement of non-qualified assistants in providing minor ailments services or acting as the first point of contact. Dependent on the nature of the request, this involvement could impact on patient safety and quality of pharmacy services. The quote below clearly indicates this engagement should involve a pharmacist or pharmacy technician:


*“Non-qualified assistants are sometimes involved in providing pharmacy services to patients presenting with minor ailments, but they shall consult the pharmacist or pharmacy technicians for service provision.”*
(Pharm1)

It appeared that the issues of non-qualified assistants’ involvement in the MMA are the result of the COVID-19 pandemic and pharmacist absence from the pharmacy, particularly during pharmacy peak opening hours. It was evident that pharmacist respondents were aware of the regulations regarding non-qualified assistants’ involvement in providing minor ailment services, indicating this is inappropriate practice:


*“They (non-qualified assistants) cannot provide such service. The law has stipulated that they are not supposed to provide the service. However, during the peak hours at the pharmacy, non-qualified assistants (who collects the prescriptions or deliver medicines) will come to help us…”*
(Pharm3)

#### 3.1.3. Inconsistent Education and Training of Pharmacy and Pharmacy Technician Students

Inconsistencies in the MMA education and training may establish how pharmacy technicians may take a larger role in the MMA services, which disrupts the Agency Theory model relationships. Further, there was a perception among pharmacist and pharmacy technician respondents that pharmacy education and training delivered by universities varied and lacked relevance to pharmacy practice. Minor ailments topics were often integrated into various therapeutic disciplines:


*“Back then, the lectures at the university only provided us with theoretical explanation without going to further detail regarding what we can experience during the practice…there was no special course on minor ailments since the materials related to minor ailments were covered in some units such as Pharmacology and Pharmaceutical Science. Apart from that, there is a wide gap between theoretical aspects and practical aspects.”*
(Pharm9)

The lack of specific courses on minor ailments, which included practical aspects, has led to inadequate preparation of pharmacists to manage minor ailments on graduation. Further, pharmacist respondents stated that including the concepts of MMA into other courses or subjects widened the gap between pharmacy education and practice.

Consequently, the levels of capability reported in providing minor ailments services varied as to when the pharmacy students graduated. Several pharmacists lacked the confidence to provide minor ailment services when they initially graduated due to a lack of integrated training received at university. They had turned to self-learning to improve their competence.


*“I was not confident back then, because I had to learn everything from scratch (when graduated).”*
(Pharm6)


*“I believe that confidence is more closely related to experience…Some universities may provide no clinical placement program for the undergraduate level…”*
(Pharm2)

Pharmacy training received much attention from pharmacist and pharmacy technician respondents as they viewed it as essential to improving the standard of pharmacy practice. There was a mixed response from pharmacist respondents regarding the format of the proposed training. Some pharmacists proposed clinical practice or hands-on experiential training. Others proposed a longer duration of clinical placement during the Apothecary (final year) program. The lack of standardised and coordinated training was reported by the majority of pharmacist and pharmacy technician respondents as the main barrier to providing appropriate MMA, and thus, they would like coordinated training in this area.

Respondents who proposed hands-on experiential training during the pharmacy course stated their opinions as follows: 


*“…there should be some kinds of training or seminars and internships with hands-on experience with patients because a mere theoretical practice will not prepare students with the real practices.”*
(Pharm3)


*“…it is necessary to add a clinical placement at a community pharmacy, because in the past, clinical placements were only conducted in hospitals or in industries.”*
(Tech7)

While pharmacists who preferred a longer duration of clinical placement and practical training (internship) after the Apothecary program stated that such activities would provide students with a better learning experience. Further, there was a recommendation that there was a need to add a clinical placement in the Bachelor of Pharmacy program prior to undertaking the Apothecary program:


*“…clinical placement would be better if the duration was extended….ideally there should be a longer duration for fieldwork practice.”*
(Pharm4)


*“….clinical placement should be made available at the undergraduate level and if possible, the duration should be increased to at least three months or one semester.”*
(Pharm2)

### 3.2. Theme Two: Lack of Understanding of the Scopes of Practice of Pharmacists and Pharmacy Technicians

In the community pharmacy setting, the pharmacist’s role in managing minor ailments was supported by pharmacy technicians. Pharmacy technicians may perform many pharmacy services. However, they should know when to refer to a pharmacist. Unfortunately, there was lack of consensus between the pharmacists and pharmacy technicians with regards to each other’s scope of practice.

#### 3.2.1. No Clear Demarcation between the Scope of Practice of Pharmacists and Pharmacy Technicians

Although pharmacy technicians may legally perform many pharmacy services, it is considered that they should have been educated and trained regarding when to refer a minor ailment to a pharmacist (beyond their scope). There was a lack of consensus between the pharmacists and pharmacy technicians with regards to each other’s scope of practice. The interviews suggested no clear limits existed in the scopes of practice between pharmacists and pharmacy technicians: 


*“There is no such difference (scope of practice) at my pharmacy. But there should be a difference. Patients are mostly served by both pharmacists and pharmacy technicians.”*
(Pharm1)


*“Since I work in this pharmacy, there is no strict difference between the service provision for minor ailments that should be handled by a pharmacy technician or by a pharmacist. From my experience, the complaints of minor ailments are all the same…”*
(Tech2)

#### 3.2.2. Inconsistencies in When to Refer Serious Minor Ailments to a Doctor

In the case of MMA, many patients who come to the pharmacy will be advised and treated without a referral to another healthcare professional. However, there will be some potentially serious ailments that require appropriate treatment by a doctor. Many respondents indicated inconsistencies in which ailments should be referred: 


*“…I shall refer some patients who suffer from shortness of breath and heartburn, complaints of pain in the left chest to see a doctor. In the case of a swollen leg, I will suggest the patient see the doctor.”*
(Pharm9)


*“Whenever we have patients with complaints of diabetes and refuse to go to the doctor, I usually check their sugar levels and ask if they have shortness of breath. If the patient is only taking metformin, I suggest that they change their medicine because it did not downgrade their sugar levels.”*
(Tech3)

### 3.3. Theme Three: Provision of Prescription-Only Medicines for MMA

Similarity occurred between the pharmacist and pharmacy technician respondents that non-prescription supply of prescription-only medicines was a common practice in community pharmacies in Central Java. The interviews indicated that many pharmacists and pharmacy technicians had dispensed all classifications of medicines, including prescription-only medicines for treating minor ailments. The interview analysis showed that the Indonesian health authorities have not initiated any program to address concerns regarding non-prescription supply of prescription-only medicines, which has resulted in this becoming common practice.

#### 3.3.1. Commonly Purchased Medicines, including the Non-Prescribed Sales of Antibiotics and Its Existing Regulations

Both respondent groups raised the illegal supply of prescription-only medicines without a valid prescription being a common practice in community pharmacies in Central Java:


*“…even though the patient came without a prescription, whenever their blood pressure turned out to be high, and they had never taken any medicine, usually I would give them a common blood pressure medicine such as amlodipine *.” * Labelled as prescription-only medicines.*
(Tech8)

Some pharmacists indicated that antibiotics could be easily purchased without a prescription from pharmacy and non-pharmacy outlets, and thus, considered it a common practice. 


*“…we do serve to patients because antibiotics are affordable. You can get amoxicillin only for Rp.4500 (USD 0.30). Even you can have it only for Rp.2000 (USD 0.15).”*
(Pharm12)

Patients were reportedly purchasing antibiotics without a prescription when they brought a sample container of the medicines to the pharmacy: 


*“Yes, if they bring a sample of the medicine (antibiotics), they can easily get the medicine. Bringing a sample of the medicine means that they have used the medicine before.”*
(Tech6)

#### 3.3.2. Source of Non-Prescribed Sale of Antibiotics

The results of this study also suggested that non-pharmacy healthcare professionals were one of the sources of non-prescribed sale of antibiotics. The interviews reflected a lack of adherence to the current regulations and inconsistent monitoring by the Indonesian authorities:


*“In practice, even though my pharmacy does not sell antibiotics freely (supplied), it turns out that some midwives and nurses uncontrollably provide patients with antibiotics. They keep on providing patients with antibiotics…”*
(Pharm4)

Respondents identified that unauthorised supplies of antibiotics were made routinely by non-pharmacy healthcare professionals from a range of outlets, which raised concerns about the ease of access the community has to antibiotics from multiple sources rather than just most community pharmacies (there were a few pharmacies that refused to sell antibiotics).

Apart from the illegal dispensing of antibiotics, one pharmacist also reported that other healthcare professionals also engaged in the illegal provision of other prescription-only medicines:


*“I saw some midwives and mantri (orderlies) providing some medicines that were out of their scope, and sometimes they gave the wrong medicine. Even worse, they also gave the patient who had a cardiac history with the medicine that contradicted the illness…”*
(Pharm12)

#### 3.3.3. Patients Misconceptions about Antibiotics

Consumers’ lack of knowledge or misconceptions about antibiotics were commonly raised by respondents in this study. Both pharmacists and pharmacy technicians described that their patients demanded antibiotics indiscriminately for any minor ailment condition. Patients believed that antibiotics were “powerful” medicines that can effectively treat minor ailments such as common cold, cough, diarrhoea, infections, itch, wounds, and pain:

*“A common example that allows the provision of antibiotics is urinary tract infection…patients prefer to come directly to the pharmacy rather than check with the doctor. Pharmacists will usually suggest increasing broader spectrum of activity, such as giving urinary Urotractin * antibiotics.” * Antibiotic contains pipemidic acid*.(Tech8)

Both pharmacist and pharmacy technician respondents indicated that patients pressured them to provide antibiotics without a prescription:


*“Patients will surely become upset. Instead, they will try to obtain antibiotics in other pharmacies. In their view, why cannot they have them at my pharmacy?”*
(Pharm12)


*“I have encountered some obstacles like the provision of antibiotics, as previously mentioned. If I do not give it to the patients, they remain nagging about it.”*
(Tech12)

#### 3.3.4. Pharmacy Technicians’ Commercial Interest Pushing the Sale of Non-Prescribed Antibiotics

Pharmacy technician respondents stressed their concerns regarding “patient demand” as an influencing factor for the non-prescribed sale of antibiotics. They stated that they dispensed antibiotics without a prescription because of a culture of not refusing any customer request for medicines:


*“The poor condition of the pharmacy makes it unable to receive many prescriptions. Thus, the pharmacy had to sell antibiotics easily, and sell all medicines, both prescription-only medicines and antibiotics for profit.”*
(Tech5)


*“I know that antibiotics should not be traded easily. Nonetheless, if we do not sell antibiotics, the pharmacy will not make a profit.”*
(Tech6)

#### 3.3.5. Weak Enforcement of Regulations

Although several pharmacist respondents indicated they were aware of the regulations to restrict the supply of prescription-only medications, including antibiotics without prescription, this inappropriate practice continues due to a lack of enforcement of the regulations by Indonesian authorities. The following quotes illustrate this:


*“BPOM (the Indonesian National Agency of Food and Drug Control) only checks medicines without a logo. BPOM never asked about antibiotics in detail.”*
(Pharm9)


*“Each representative from the pharmacy was invited to come (to BPOM office) to have the briefing (about dispensing antibiotics). However, it remains merely a briefing and direction without any strict supervision.”*
(Tech6)

### 3.4. Agency Theory Application to the Provision of MMA

Principals (community pharmacy patients) have delegated decision-making to a pharmacist or pharmacy technician to assist them with their minor ailments. Where a patient considers they need assistance with the management of a minor ailment, they would seek the expertise of a pharmacist or a pharmacy technician. A model to describe these relationships in the current Indonesian legal context is shown in Figure 1. Agency Theory defines that the longer the principal and agent relationship exists, the more the principal learns about their agent [30]. Patients would be seeking to establish a clinical relationship with an appropriate pharmacist and/or pharmacy technician. The findings in this study describe two principal–agency relationships: between the principal (pharmacy patients) and the pharmacy staff; and between pharmacy technicians and pharmacists. Regarding the first relationship, a person comes to the pharmacy to purchase a medicine and with the exception of ensuring they know how to use it, no agency relationship is established as it was a transactional occurrence only. The second occurrence is when a patient seeks advice about a minor ailment and is directed to the pharmacy technician or pharmacist, or may require a referral from the pharmacy technician to the pharmacist to manage more complex ailments. This scenario is developed in Figure 1. This establishes a direct agency relationship between the patient and a pharmacy technician or pharmacist. If the ailment is beyond the scope of a pharmacist, the principal would be referred by the pharmacist to an appropriate health professional. Applying this model to the main themes from this study: (i) Inconsistencies in MMA practice causes disruption of these relationships due to a lack of standard procedures in a pharmacy, or the absence of a pharmacist causing a secondary relationship to be established with a pharmacy technician; (ii) The principal would be unclear regarding the scopes of practice of pharmacists and pharmacy technicians retarding the development of an understanding of their capabilities to provide MMA. It is the professional responsibility of the agent to not provide services beyond their competence (scope of practice). This can include managing ailments beyond their scope of practice, potentially leading to a breakdown of the relationship; (iii) Regarding the provision of prescription-only medicines, including antibiotics, by pharmacists and pharmacy technicians has shifted their practice towards a more medical role that is beyond both their scopes of legal practice. On the other hand, the relationship between patient–doctor has been disrupted by the fact that patients are engaging directly with the pharmacist or pharmacy technician (when supplying prescription-only medicines). This has gradually established an improper relationship that has caused the principal (patient) to demand the agent (pharmacy staff) to provide antibiotics and other medicines for chronic diseases that should be provided only on the presentation of a prescription. Many pharmacists manage this as a repeat supply of a previously prescribed medicine and therefore safe for the patient. It is, however, illegal to supply these medicines without authorisation from a prescriber who can legally prescribe the medicine. Legally, a patient would establish an agency relationship with the doctor and a secondary relationship with a pharmacist (or pharmacy technician) who dispenses the prescription and any subsequent repeats over time. This secondary relationship is also disrupted if the doctor dispenses the medicine.

## 4. Discussion

This is the first study exploring pharmacists’ and pharmacy technicians’ perceived scopes of practice to deliver minor ailment services in community pharmacies within the context of Agency Theory in community pharmacies in Indonesia. A model, adopted from the Agency Theory, provides insight for the MMA provision in community pharmacies evident from the interview themes, as shown in Figure 1. The Agency Theory model applies when the “principal” (patients) depends on second parties (the pharmacist or pharmacy technician) to perform actions on their behalf, as they are expected to have the specific expertise to provide the service or will refer a patient to a suitable source of expertise [32]. 

A major theme arising from this study was inconsistencies in MMA practice, related to several factors. In many pharmacies, pharmacists were not in attendance for all opening hours. The issue of pharmacist absence was previously recorded in Indonesian community pharmacies, where only 14–25% of pharmacists work on a full-time basis and otherwise delegate their professional work to pharmacy assistants [18,33,34,35]. Community pharmacies in Indonesia are commonly open for extended hours, making it difficult for the primary pharmacist to be in attendance for the entire opening period (one pharmacist usually covers one shift). The concept of employing pharmacists to cover the remaining time has occurred inconsistently. Consequently, professional pharmacy services were often delegated to other pharmacy staff (i.e., pharmacy technicians or non-qualified staff: non-qualified staff are employees without pharmacy qualifications (academically not graduated as a pharmacy student or a pharmacy technician student) who normally collect the prescriptions or deliver medicines. These staff are available in the pharmacy to assist qualified staff by initially serving clients and completing transactions for some non-medicine or straightforward medicine requests. Non-qualified assistants in Indonesia are not allowed to provide medicines and counselling, which legally are only delivered by pharmacists and pharmacy technicians [36]. However, the findings from the interviews showed that minor ailments were being managed not only by pharmacists and pharmacy technicians, but on occasions by non-qualified assistants. This has occurred in some cases due to pharmacists’ absence from the pharmacy during some of its opening hours [37]. The lack of enforcement of current regulations contributes to the current haphazard management of MMA in community pharmacies. Pharmacists’ absence is in contravention of current regulations.

Although community pharmacists hold much higher qualifications to provide MMA, this study shows the establishment of a professional relationship with their patients is interrupted by their absenteeism and complicated by pharmacy technicians, considering they can provide an equivalent service. No staff member should carry out a service that is not within the scope of their practice. To date, there is no published data on the levels of competence of the two groups in the provision of MMA. The current situation has become embedded into routine practice, but these findings indicate it needs to be questioned by the government and pharmacy professional bodies, as the level of MMA services provided by pharmacy technicians may not be equivalent to that of a pharmacist. MMA provision to the Indonesian community is important, as no other source of minor ailment management is available. Illnesses requiring treatment by doctors can be managed in both the public and private sectors, but minor ailments are managed by the consumer alone or in conjunction with a pharmacist or pharmacy technician at a community pharmacy. This forms an essential element of the primary care system. Currently, the pharmacy technician may be usurping the role of the pharmacist in MMA, potentially leading to poorer health outcomes for the general public. 

Both respondent groups in this study indicated that university training in the MMA was variable across pharmacy teaching institutions. Besides pharmacy training, this study also found that MMA subject/unit topics were of variable length, relevance, and the amount of experiential learning was varied, as in many countries [38,39,40]. Professional organisations should review current curricular and make submissions to facilitate the setting of appropriate accreditation standards, which all teaching universities must comply.

This study has further identified a lack of clear scopes of practice for MMA, as currently perceived by community pharmacists and pharmacy technicians. This supports our findings from previous studies [37,41]. One of these studies indicated that pharmacy technicians considered the scope of minor ailments they could manage was much broader than that perceived by pharmacist colleagues [37]; this is not, however, well supported from the interview findings, where little difference in scopes of practice was often evident in regard to which group managed minor ailments.

A limited number of “Pharmacist-Only” medicines are registered specifically for the MMA. Although, legally, they define a scope of practice differential for pharmacists with respect to pharmacy technicians, the current provision of prescription medicines by both groups currently renders “Pharmacist-Only” medicines irrelevant. 

The Agency Theory is also confused when repeat prescription-only medicines are provided for managing chronic diseases in community pharmacies without medical authorisation. The relationship between the doctor and the patient is disrupted when the community pharmacist becomes the first point of contact for the patient with chronic diseases [42]. Normally, a patient would delegate their authority to the pharmacist to provide repeat medicines based on a medical prescription. However, commonly, the patient is providing a labelled empty container to the pharmacy and placing their trust in the pharmacist to fulfil this role, without consultation with their doctor. These results have some similarities with studies conducted in some developing countries such as Thailand, Malaysia and Vietnam [43,44,45,46].

The request from consumers/patients for a broader range of medicines for minor ailments, including antibiotics and prescription-only medicines for chronic diseases, can be described by the Agency Theory model, where the patient has delegated their authority to the community pharmacist rather than the doctor (particularly for chronic diseases). Consumers then rely more heavily on the pharmacist rather than the doctor as their preference to seek advice. In this case, the pharmacist’s position is shifted in the Agency Theory and they take on a pseudo medical role, beyond their scope of practice [30]. Further, the patients’ demands for antibiotics identifies their lack of understanding regarding the appropriate use of antibiotics and that their indiscriminate use can lead to antibiotic resistance [47].

This study has highlighted a need for the professional bodies to develop practice guidelines for the MMA and achieve consensus on the scopes of practice of pharmacists and pharmacy technicians to assist in reducing current inconsistencies in practice. Standardisation of practice aligned with consistent and relevant MMA education of pharmacists and pharmacy technicians would ultimately improve patient safety. Limitations in this research include the fact that the participants’ experiences that were reported may be different in some other parts of Indonesia, where access to health services is more limited. Some perceptions could have been influenced by the COVID-19 pandemic that was severe in Indonesia during the period of data collection. This study only included pharmacists’ and pharmacy technicians’ perceptions, thus, further research involving other parties’ (e.g., patients, academics, government staff, professional organisations) perceptions may enrich the findings. A purposive sampling method used in this study was not intended to be generalised, thus it is a limitation of the method. Participants’ diverse experiences in the interviews may not be explored within a complete portrait of MMA services and recommendations arising from the discussion may not answer some issues. Further, purposive sampling in this study obtained a range of pharmacists’ and pharmacy technicians’ views and recruited participants from several cities in Central Java rather than representing them proportionally.

## 5. Conclusions

This study indicates that the delivery of MMA is currently inconsistent with respect to scope of practice partially predicated by a lack of practice guidelines. Patients seeking MMA services can be modelled by the Agency Theory, where the patient has delegated their authority to the community pharmacist or pharmacy technician. The pharmacist’s position is disrupted in the Agency Theory, partly because of their absence from the pharmacy. Relevant regulations and health laws should be applied to ensure pharmacists and pharmacy technicians practise within their respective scopes. Further, discrepancies and inconsistencies exist in the education and training for the MMA for pharmacists and pharmacy technicians in Indonesia, which needs review.

## Figures and Tables

**Figure 1 pharmacy-11-00132-f001:**
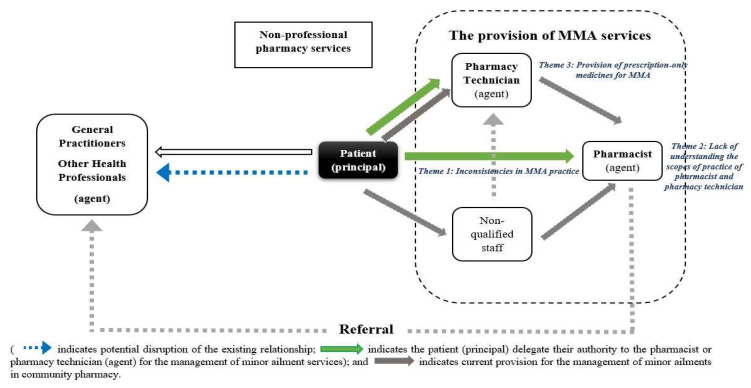
Model of the current management of minor ailments (MMA) provision.

**Table 1 pharmacy-11-00132-t001:** Demographic characteristics of pharmacist (n = 12) and their practice settings.

Code	Gender	Age	Years of Practice	Pharmacy Type	Position	Average Working Hours Per Week	Average Consumers Per Week	Average MMA Patients Per Week
Pharm1	Female	31–40	11–15	Independent	Pharmacy manager and owner	51+	251–350	>70
Pharm2	Female	31–40	<2	Independent	Pharmacy manager and owner	51+	451–550	>70
Pharm3	Female	21–30	<2	Co-located with a medical centre *	Additional pharmacist	41–50	>700	61–70
Pharm4	Female	21–30	2–5	Independent	Pharmacy manager and owner	51+	351–450	>70
Pharm5	Male	21–30	<2	Franchise	Pharmacy manager	41–50	>700	>70
Pharm6	Male	41–50	>15	Independent	Pharmacy manager and owner	41–50	551–700	>70
Pharm7	Female	21–30	2–5	Franchise	Pharmacy manager	31–40	100–150	61–70
Pharm8	Female	31–40	11–15	Independent	Pharmacy manager	41–50	<100	61–70
Pharm9	Female	41–50	6–10	Independent	Pharmacy manager	31–40	351–450	21–30
Pharm10	Female	31–40	6–10	Independent	Pharmacy manager and owner	41–50	151–250	41–50
Pharm11	Female	21–30	<2	Co-located with a doctor’s practice *	Additional pharmacist	51+	551–700	21–30
Pharm12	Male	31–40	6–10	Independent	Pharmacy manager and owner	41–50	151–250	41–50

MMA = Management of minor ailments. * Co-located with a medical centre (direct access for the public to a pharmacy).

**Table 2 pharmacy-11-00132-t002:** Demographic characteristics of pharmacy technician (n = 12) and their practice settings.

Code	Gender	Age	Years of Practice	Pharmacy Type	Pharmacy Ownership	Average Working Hours Per Week	Average Consumers Per Week	Average MMA Patients Per Week
Tech1	Male	21–30	2–5	Independent	Non-pharmacist	41–50	451–550	>70
Tech2	Female	21–30	2–5	Franchise	Non-pharmacist	41–50	>700	>70
Tech3	Female	21–30	6–10	Independent	Non-pharmacist	21–30	251–350	>70
Tech4	Female	21–30	<2	Independent	Non-pharmacist	51+	151–250	>70
Tech5	Female	21–30	6–10	Independent	Non-pharmacist	41–50	151–250	31–40
Tech6	Female	41–50	>15	Independent	Non-pharmacist	21–30	451–550	>70
Tech7	Male	21–30	2–5	Independent	Non-pharmacist	41–50	451–550	>70
Tech8	Female	21–30	2–5	Franchise	Non-pharmacist	41–50	>700	>70
Tech9	Female	21–30	<2	Independent	Non-pharmacist	41–50	551–700	>70
Tech10	Female	21–30	2–5	Franchise	State-owned enterprise	41–50	451–550	>70
Tech11	Female	21–30	2–5	Independent	Non-pharmacist	41–50	>700	>70
Tech12	Female	31–40	>15	Co-located with a medical centre *	Regional-owned enterprises	41–50	>700	>70

MMA = Management of minor ailments. * Co-located with a medical centre (direct access for the public to a pharmacy).

**Table 3 pharmacy-11-00132-t003:** Emerged themes and sub-themes in relation to the perspectives of the current management of minor ailments (MMA) practice as perceived by pharmacist (n = 12) and pharmacy technician (n = 12) respondents.

Key Theme	Theme	Sub-Theme
Perspectives of the current MMA practice	Inconsistencies in MMA practice	Pharmacists’ absence and no standard procedures were evident for the provision of minor ailment services Non-qualified assistant involvement providing basic services Inconsistent education and training in the MMA of pharmacy and pharmacy technician students
	Lack of understanding the scopes of practice of pharmacist and pharmacy technician	No clear demarcation between the scope of practice of pharmacists and pharmacy technicians Inconsistencies in when to refer serious minor ailments to a doctor
	Provision of prescription-only medicines for MMA	Commonly purchased medicines, including non-prescribed sale of antibiotics and lack of pharmacy services regulations Source of non-prescribed sale of antibiotics Patients’ misconceptions about antibiotics Pharmacy technicians commercial interest pushing the sale of non-prescribed antibiotics * Weak enforcement of regulations

* Sub-theme emerged from the pharmacy technicians’ interviews.

## Data Availability

Data are available upon request.

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
