# Peer review of "Pharmacists and Pharmacy Technicians’ Perceptions of Scopes of Practice Employing Agency Theory in the Management of Minor Ailments in Central Indonesian Community Pharmacies: A Qualitative Study"

_pharmacy, 2023, doi:10.3390/pharmacy11050132_

Round 1

Reviewer 1 Report

I appreciate the opportunity to review the paper. In my opinion, qualitative studies are underused in assessing pharmacy practice. The opinion about practice received directly from pharmacy professionals is essential in planning and managing pharmacy services and pharmaceutical education and training. 

Reviewer 2 Report

I would like to extend my sincere appreciation to the authors for allowing me to review and provide feedback on their insightful manuscript.

The study investigates the viewpoints of pharmacists and pharmacy technicians in Central Java, Indonesia, regarding the scope of practice for minor ailment services in community pharmacies. The paper aims to comprehend the relationships between healthcare professionals and patients within this context by employing Agency Theory. Utilizing a qualitative approach, the study explores the perceptions of pharmacists and pharmacy technicians concerning the scope of practice for minor ailment services in community pharmacies in Central Java, Indonesia. Additionally, the paper references previous research conducted in other countries, such as Australia, which examined the education and training of pharmacists and pharmacy technicians. Ultimately, the study emphasizes the necessity for well-defined guidelines and regulations to ensure the consistent and safe delivery of minor ailment services in community pharmacies in Central Java, Indonesia.

Major points

1.       Could the authors please describe in further detail the purposive sampling method that was employed? I am unsure between “criterion sampling” and “maximum variation sampling”.

2.       Could the authors please provide a figure of the coding tree (or trees, as there were two samples)?

3.       Could the authors describe further the limitations of this paper, for example: on the perspectives of patients or other healthcare professionals; on the impact of cultural or social factors on the perceptions of pharmacists and pharmacy technicians; as the authors did not provide a quantitative analysis of the data this may limit the generalizability of the findings.

4.       Could the authors please discuss the non-qualified assistants' involvement in providing pharmacy services to patients presenting with minor ailments? The situation should be compared with similar cases/situations in other countries.

Minor points

1.       There is a repetition of some phrases in the Methods section (lines 88 – 96  and lines 115 – 123)

Reviewer 3 Report

Nicely done study, well written and easy to follow. Only minor suggestions for the methods section. 

The second paragraph of the ethical approval section isn’t about ethical approval, it is about the selection of participants and how the data were collected. The paragraph provides an incomplete picture of the selection and data collection process – therefore it is recommended to incorporate the information into the appropriate sub-sections later on in the methods section. (i.e., 2.3 and 2.4).

In section 2.2, include that the study was conducted virtually via online interviews. Include what software was used to connect with participants.

Be explicit in what the purposeful sampling criteria were.

Were individuals randomly selected from the acquired list? Did you avoid contacting pharmacists and pharmacy techs from the same pharmacy? How were individuals contacted? Email? Phone? Etc.? what were the dates of the data collection?

No issues

Reviewer 4 Report

The presented paper is interesting and may be a valuable source for readers of the Journal. I have, however, several comments – please find them below:

-      -  a fragment is repeating in the Methods section – lines 88-96 and 115-123 say exactly the same thing

-     -   in Methods section text mentions thematic analysis (line 163) but the reference [28] is on thematic content analysis – these are two different approaches and coding framework for example is incoherent with thematic analysis – name of the method should be changed to thematic content analysis in line 163 as in the cited reference

-     -   line 193 – did saturation occur at 9th interview for pharmacists and also 9th for technicians?

-     -   Themes begin already with subthemes without any introductory or explanatory fragment. At least a few sentences should be added at the beginning of each theme to introduce the theme, the division into subthemes and show/justify their relation and inclusion into a given theme – now it is not always clear

-     -   line 487 - 3.4. Agency Theory application to the provision of MMA – I wonder if this fragment should not be a part of the Discussion as these are not ‘results’ in a pure sense. However, this is only a suggestion not something I feel particularly strongly about.

-    -    I enjoyed the Figure 1. However, maybe it would be even more explanatory if subthemes were included

-      -  I struggle to understand who exactly are these non-qualified assistants – for researchers from other countries where other regulations are present (and enforced by the government) such a situation may be difficult to understand – I mean who are they and how they gained access to the pharmacy?
